# Metabolite profiling of human renal cell carcinoma reveals tissue-origin dominance in nutrient availability

Keene L Abbott[1,2,3†], Ahmed Ali[2,3†], Bradley I Reinfeld[4,5,6†], Amy Deik[3], Sonu Subudhi[7], Madelyn D Landis[5], Rachel A Hongo[5], Kirsten L Young[5], Tenzin Kunchok[8], Christopher S Nabel[2,9,10], Kayla D Crowder[8], Johnathan R Kent[11], Maria Lucia L Madariaga[11], Rakesh K Jain[7], Kathryn E Beckermann[5], Caroline A Lewis[8‡], Clary B Clish[3], Alexander Muir[12], W Kimryn Rathmell[5,13], Jeffrey Rathmell[13,14]*, Matthew G Vander Heiden[1,2,3,15]*

[1]Department of Biology, Massachusetts Institute of Technology, Cambridge, United States; [2]Koch Institute for Integrative Cancer Research, Massachusetts Institute of Technology, Cambridge, United States; [3]Broad Institute of MIT and Harvard, Cambridge, United States; [4]Medical Scientist Training Program, Vanderbilt University, Nashville, United States; [5]Department of Medicine, Vanderbilt University Medical Center (VUMC), Nashville, United States; [6]Graduate Program in Cancer Biology, Vanderbilt University, Nashville, United States; [7]Steele Laboratories of Tumor Biology, Department of Radiation Oncology, Massachusetts General Hospital and Harvard Medical School, Boston, United States; [8]Whitehead Institute for Biomedical Research, Cambridge, United States; [9]Department of Medicine, Massachusetts General Hospital, Boston, United States; [10]Harvard Medical School, Boston, United States; [11]Department of Surgery, University of Chicago Medicine, Chicago, United States; [12]Ben May Department of Cancer Research, University of Chicago, Chicago, United States; [13]Vanderbilt Center for Immunobiology and Vanderbilt-Ingram Cancer Center, VUMC, Nashville, United States; [14]Department of Pathology, Microbiology and Immunology, VUMC, Nashville, United States; [15]Dana-Farber Cancer Institute, Boston, United States

*For correspondence:
jeff.rathmell@vumc.org (JR);
mvh@mit.edu (MGVH)

†These authors contributed equally to this work

Present address: ‡UMass Chan Medical School, Program in Molecular Medicine, Worcester, United States

**Abstract** The tumor microenvironment is a determinant of cancer progression and therapeutic efficacy, with nutrient availability playing an important role. Although it is established that the local abundance of specific nutrients defines the metabolic parameters for tumor growth, the factors guiding nutrient availability in tumor compared to normal tissue and blood remain poorly understood. To define these factors in renal cell carcinoma (RCC), we performed quantitative metabolomic and comprehensive lipidomic analyses of tumor interstitial fluid (TIF), adjacent normal kidney interstitial fluid (KIF), and plasma samples collected from patients. TIF nutrient composition closely resembles KIF, suggesting that tissue-specific factors unrelated to the presence of cancer exert a stronger influence on nutrient levels than tumor-driven alterations. Notably, select metabolite changes consistent with known features of RCC metabolism are found in RCC TIF, while glucose levels in TIF are not depleted to levels that are lower than those found in KIF. These findings inform tissue nutrient dynamics in RCC, highlighting a dominant role of non-cancer-driven tissue factors in shaping nutrient availability in these tumors.

## eLife assessment

This study provides an **important** finding that the local abundance of metabolites impacts the biology of the tumor microenvironment by utilizing kidney tumors from patients and adjacent normal tissues. The evidence supporting the claims of the authors is **convincing**. The work will be of interest to the research community working on metabolism and kidney cancer especially.

## Introduction

Nutrient availability within the tumor microenvironment (TME) can influence cancer progression, therapeutic response, and metastasis (*Muir and Vander Heiden, 2018*; *Muir et al., 2017*; *Abbott et al., 2023*; *Cantor et al., 2017*; *Vande Voorde et al., 2019*; *Davidson et al., 2016*; *Gui et al., 2016*; *Ferraro et al., 2021*; *Jin et al., 2020*; *Ngo et al., 2020*; *Faubert et al., 2020*; *Rossiter et al., 2021*). How metabolite availability is regulated in the TME has been a topic of extensive research (*Apiz Saab and Muir, 2023*; *Sullivan et al., 2019*; *Lyssiotis and Kimmelman, 2017*; *Sullivan and Vander Heiden, 2019*; *Reinfeld et al., 2021*); however, a comprehensive analysis of what nutrients are found in cancer tissue interstitial fluid, and how this deviates from nutrients available in the corresponding normal tissue, has not been conducted. Thus, whether global tumor nutrient availability is actively modified by tumor cells, the accompanying immune, endothelial, and stromal cells, or predominantly influenced by the conditions of the originating resident tissue, is not known. Better understanding the determinants of nutrient availability in the TME will inform efforts to understand preferences for specific cancers to develop in specific tissues (*Faubert et al., 2020*; *Bergers and Fendt, 2021*; *Mosier et al., 2021*), and could enable better matching of patients with cancer therapies whose efficacy is influenced by what nutrients are present in tumor tissue (*Muir and Vander Heiden, 2018*; *Muir et al., 2017*; *Abbott et al., 2023*; *Cantor et al., 2017*; *Vande Voorde et al., 2019*; *Rossiter et al., 2021*).

Surgical management of human renal cell carcinoma (RCC) allows for the simultaneous sampling of tumor interstitial fluid (TIF), adjacent healthy kidney tissue interstitial fluid, and plasma collected from patients undergoing nephrectomy. This afforded the opportunity to assess metabolites across these different samples and assess how nutrients found in tumor tissue relate to those found in corresponding normal tissue and blood. Notably, a number of distinct types of primary renal cancers, driven by a diverse set of genetic programs, arise from the kidney (*Linehan et al., 2010*; *Linehan, 2012*). This study was unselected with regard to tumor histology, although the diversity of tumors offered a further opportunity to examine the range of nutrient variation across kidney tumors versus normal tissue, as well as among tumors with a variety of histologies. Supported by mass spectrometry-based quantification of polar metabolites and lipids in normal and cancerous tissue interstitial fluid and plasma, we find that nutrient availability in TIF closely mirrors that of interstitial fluid in adjacent normal kidney tissue (kidney interstitial fluid, KIF), but that nutrients found in both interstitial fluid compartments differ from those found in plasma. This analysis suggests that the nutrients in kidney tissue differs from those found in blood, and that nutrients found in kidney tumors are largely dictated by factors shared with normal kidney tissue. We additionally present metabolite measurements in these conditions as a resource to support further study and modeling of the local environment of RCC and normal kidney physiology.

## Results and discussion

We isolated interstitial fluid from RCC tumors (TIF) and from adjacent non-tumor bearing kidney tissue (KIF), then performed quantitative polar metabolomics ($n$ = 104 metabolites) and lipidomics ($n$ = 210 metabolites) of these fluids alongside plasma collected from patients with kidney cancer at the time of nephrectomy (*Figure 1A*; *Supplementary files 1–3*). Principle component analysis (PCA) of polar and lipid metabolite levels measured in these samples revealed some differences between TIF and KIF, but these tissue analytes principally clustered distinctly from plasma (*Figure 1B,C*), highlighting that the nutrients found in circulation differ extensively from those found in tissue interstitial fluid. In particular, lipid features were tightly concordant between KIF and TIF, in contrast to plasma where lipid components have widespread variation.

We also performed PCA of polar and lipid metabolite levels measured in material derived from the subset of patients with clear cell RCC (ccRCC), as this is the most common subtype of RCC (*Linehan, 2012*; *Muglia and Prando, 2015*) and represented the largest fraction of the patient samples available

**eLife digest** Cancer cells convert nutrients into energy differently compared to healthy cells. This difference in metabolism allows them to grow and divide more quickly and sometimes to migrate to different areas of the body. The environment around cancer cells – known as the tumor microenvironment – contains a variety of different cells and blood vessels, which are bathed in interstitial fluid. This microenvironment provides nutrients for the cancer cells to metabolize, and therefore influences how well a tumor grows and how it might respond to treatment.

Recent advances with techniques such as mass spectrometry, which can measure the chemical composition of a substance, have allowed scientists to measure nutrient levels in the tumor microenvironments of mice. However, it has been more difficult to conduct such studies in humans, as well as to compare the tumor microenvironment to the healthy tissue the tumors arose from.

Abbott, Ali, Reinfeld et al. aimed to fill this gap in knowledge by using mass spectrometry to measure the nutrient levels in the tumor microenvironment of 55 patients undergoing surgery to remove kidney tumors. Comparing the type and levels of nutrients in the tumor interstitial fluid, the neighboring healthy kidney and the blood showed that nutrients in the tumor and healthy kidney were more similar to each other than those in the blood. For example, both the tumor and healthy kidney interstitial fluids contained less glucose than the blood. However, the difference between nutrient composition in the tumor and healthy kidney interstitial fluids was insignificant, suggesting that the healthy kidney and its tumor share a similar environment.

Taken together, the findings indicate that kidney cancer cells must adapt to the nutrients available in the kidney, rather than changing what nutrients are available in the tissue. Future studies will be required to investigate whether this finding also applies to other types of cancer. A better understanding of how cancer cells adapt to their environments may aid the development of drugs that aim to disrupt the metabolism of tumors.

for analysis. We found similar clustering patterns for ccRCC metabolite levels (*Figure 1—figure supplement 1A,B*) as those found when we analyzed the data collected from all patient-derived material. Further analysis by *t*- and chi-squared tests revealed that levels of most polar metabolites do not significantly differ between TIF and KIF (*Figure 1D*, *Figure 1—figure supplement 1C*). These same analyses revealed a difference in the number of statistically different metabolites when comparing TIF and KIF lipid composition, but these differences were fewer than when comparing the lipid composition of either TIF or KIF compartments to plasma (*Figure 1E*, *Figure 1—figure supplement 1D*). Taken together, these data suggest that metabolites found in TIF and KIF are largely similar based on these measurable analytes, although there are individual polar and lipid metabolites that differ between TIF and KIF (*Figure 2A,B*, *Figure 2—figure supplement 1A,B*). These data support the notion that the human RCC tumor nutrient environment is primarily driven by factors shared with kidney tissue, rather than being extensively modified by cancer cells or other factors unique to the tumor.

Focusing on differences between TIF and KIF using data derived from the entire patient dataset, we found several metabolites to be elevated in TIF compared to KIF that agree with known metabolic characteristics of RCC. These include increased levels of 2-hydroxyglutarate (*Shim et al., 2014*; *Wang et al., 2021*), kynurenine (*Wettersten et al., 2015*; *Ganti et al., 2012*; *Lucarelli et al., 2017*), and glutathione (*Wettersten et al., 2015*; *Hakimi et al., 2016*; *Figure 2C*), all consistent with prior reports and known biology of RCC. Of note, we observed depletion of glucose and elevation of lactate in TIF compared to plasma, again consistent with known increases in glucose fermentation to lactate in this tumor type (*Kaushik et al., 2022*; *Masson and Ratcliffe, 2014*; *Kaelin, 2008*; *Courtney et al., 2018*; *Sanchez and Simon, 2018*; *Figure 2D*). However, the levels of glucose and lactate measured in TIF were similar to those measured in KIF, arguing that glucose was not further depleted in TIF compared to KIF (*Figure 2D*). These data challenge the notion that cells in the TME are necessarily glucose starved (*Burgess and Sylven, 1962*; *Gullino et al., 1964*; *Hirayama et al., 2009*; *Urasaki et al., 2012*; *Ho et al., 2015*), as the average glucose concentration is near or above the threshold identified as limiting for proliferation in a range of cancer cell lines (*Birsoy et al., 2014*). It is important to note that the levels of glucose measured in bulk TIF may not reflect local depletion in subregions of the tumor. Regardless, these findings fit with other data from RCC, as well as from other tumor

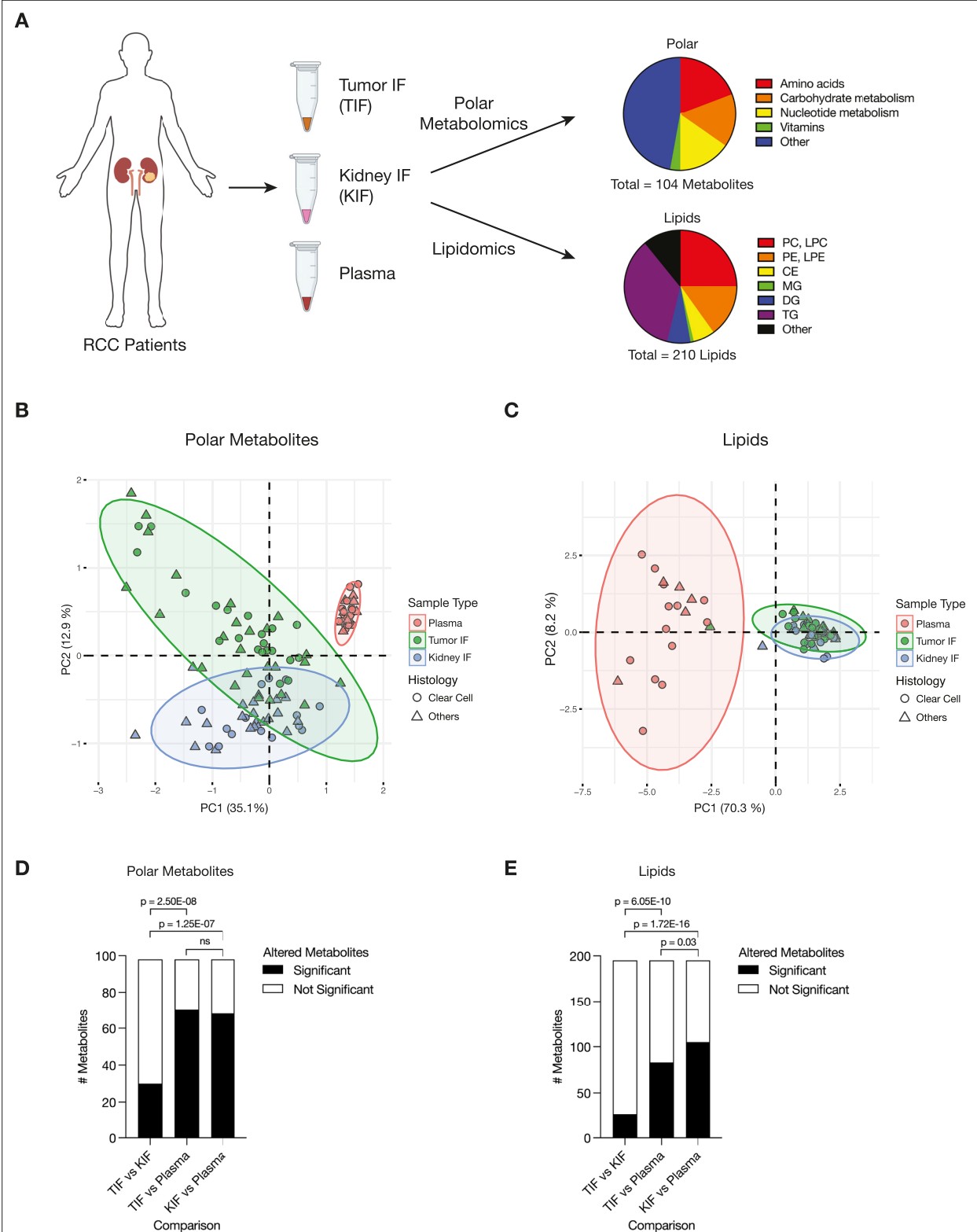

**Figure 1.** Levels of metabolites in renal cell carcinoma (RCC) tumor interstitial fluid (TIF) are similar to those found in normal kidney interstitial fluid (KIF). (**A**) Schematic depicting study design whereby samples collected from patients with RCC undergoing nephrectomy were used to derive TIF, KIF, and plasma. Samples were then subjected to polar metabolomics and lipidomics analyses. See *Supplementary file 1* for patient information, and *Supplementary file 2* for metabolite concentrations. (**B**) Principal component analysis of polar metabolites measured from the indicated RCC patient samples (*n* = 55 patients). For each sample, absolute levels of 98 polar metabolites were quantified by liquid chromatography/mass spectrometry (LC/

*Figure 1 continued on next page*

*Figure 1 continued*

MS). Data represent 55 TIF, 46 KIF, and 27 plasma samples. The 95% confidence interval is displayed. (**C**) Principal component analysis of lipid species measured from the indicated RCC patient samples (*n* = 38 patients). For each sample, relative levels of 195 lipids were assessed by LC/MS. Data represent 34 TIF, 25 KIF, and 18 plasma samples. The 95% confidence interval is displayed. (**D**) *T*-test analysis of polar metabolites (*n* = 98) that do or do not significantly differ in concentration between each site from all RCC patient samples (*n* = 55 patients). Cutoffs of |log$_2$ fold change| >1 and adjusted p-value (false discovery rate-corrected) <0.05 were used to determine significant metabolites. p-values in the plot are derived from chi-squared statistical analysis. (**E**) *T*-test analysis of lipids (*n* = 195) that do or do not significantly differ in concentration between each site from all RCC patient samples (*n* = 38 patients). Cutoffs of |log$_2$ fold change| >1 and adjusted p-value (false discovery rate-corrected) <0.05 were used to determine significant metabolites. p-values in the plot are derived from chi-squared statistical analysis. Panel A created with BioRender.com, and published using a CC BY-NC-ND license with permission.

The online version of this article includes the following figure supplement(s) for figure 1:

**Figure supplement 1.** Levels of metabolites in clear cell renal cell carcinoma (RCC) interstitial fluid are similar to those found in normal kidney interstitial fluid (KIF).

types, demonstrating that while glucose availability is reduced in the TME relative to blood, it is not completely depleted (*Sullivan et al., 2019*; *Reinfeld et al., 2021*; *Siska et al., 2017*). This finding is somewhat unexpected, given that VHL loss in ccRCC leads to increased HIF1α-driven expression of glucose transporters and glycolytic enzymes (*Linehan et al., 2010*; *Wettersten et al., 2015*; *Hakimi et al., 2016*; *Mandriota et al., 2002*; *Semenza, 2009*; *Rathmell et al., 2018*), but exchange of glucose between blood and interstitial fluid may be sufficient to prevent glucose depletion to levels that are less than those found in normal kidney tissue. Of note, these data do not necessarily argue glucose metabolism in tumors, or in malignant cancer cells within the tumor, is similar to that of normal tissue, only that availability of glucose across the regions of tissue sampled is similar. Indeed, evidence for increased glucose metabolism in some forms of human RCC is evident from [$^{18}$F]Fluorodeoxyglucose (FDG)-positron emission tomography scans (*Reinfeld et al., 2021*; *Hou et al., 2021*; *Lee et al., 2017*; *Wang et al., 2012*); however, FDG-glucose uptake is variable in the more common ccRCC subtye and can display regional variation (*Brooks et al., 2016*). In vivo tracing studies of isotope-labeled glucose have demonstrated distinct metabolic fluxes in ccRCC tumors compared to matched adjacent kidney tissue at the time of surgery (*Courtney et al., 2018*). Lastly, these data are notable with respect to the finding that ccRCC silences the key gluconeogenic enzyme fructose-1,6-bisphosphatase 1 (*Li et al., 2014*), as it suggests that loss of gluconeogenesis in cancer cells has minimal effect on local glucose availability.

Recent studies have found that arginine is depleted to very low levels in murine pancreatic cancer TIF (*Sullivan et al., 2019*; *Apiz Saab et al., 2023*; *Lee et al., 2023*), a phenomenon facilitated by myeloid-derived arginase activity (*Apiz Saab et al., 2023*). Of note, arginine is not significantly depleted in KIF or TIF from RCC patients compared to plasma, although levels of related urea cycle metabolites are reduced in TIF and KIF compared to plasma (*Figure 2E*). These findings suggest that myeloid cell infiltrates in RCC are not sufficient to lower arginine levels, and this observation may relate to the responsiveness of RCC to T-cell immune checkpoint blockade therapy (*Santoni et al., 2018*). These data are also notable with respect to the observation that primary RCC tumors commonly downregulate ASS1 expression and urea cycle activity (*Yoon et al., 2007*; *Perroud et al., 2009*; *Ochocki et al., 2018*; *Khare et al., 2021*), implying that the amount of arginine in KIF or TIF is sufficient to support cancer cell proliferation without the requirement for de novo arginine synthesis involving ASS1 upregulation. To this point, a recent study found that metastatic kidney cells in the lung upregulated ASS1 expression due to the lower availability of arginine in the lung compared to the kidney (*Sciacovelli et al., 2022*).

Given that distinctive lipid metabolism alterations are known to be associated with RCC (*Heravi et al., 2022*; *Gebhard et al., 1987*; *Saito et al., 2016*; *Riscal et al., 2021*; *Valera and Merino, 2011*), we probed our lipidomics dataset to investigate changes in lipid composition within RCC TIF compared to KIF and plasma. We found cholesterol levels in both TIF and KIF were notably lower when compared to plasma (*Figure 2F*), aligning with prior studies observing a roughly fivefold depletion of VLDL, LDL, and HDL cholesterol in tissue interstitial fluid versus plasma (*Vessby et al., 1987*; *Dabbagh and Frei, 1995*; *Parini et al., 2006*). Although cholesterol levels were dramatically lower

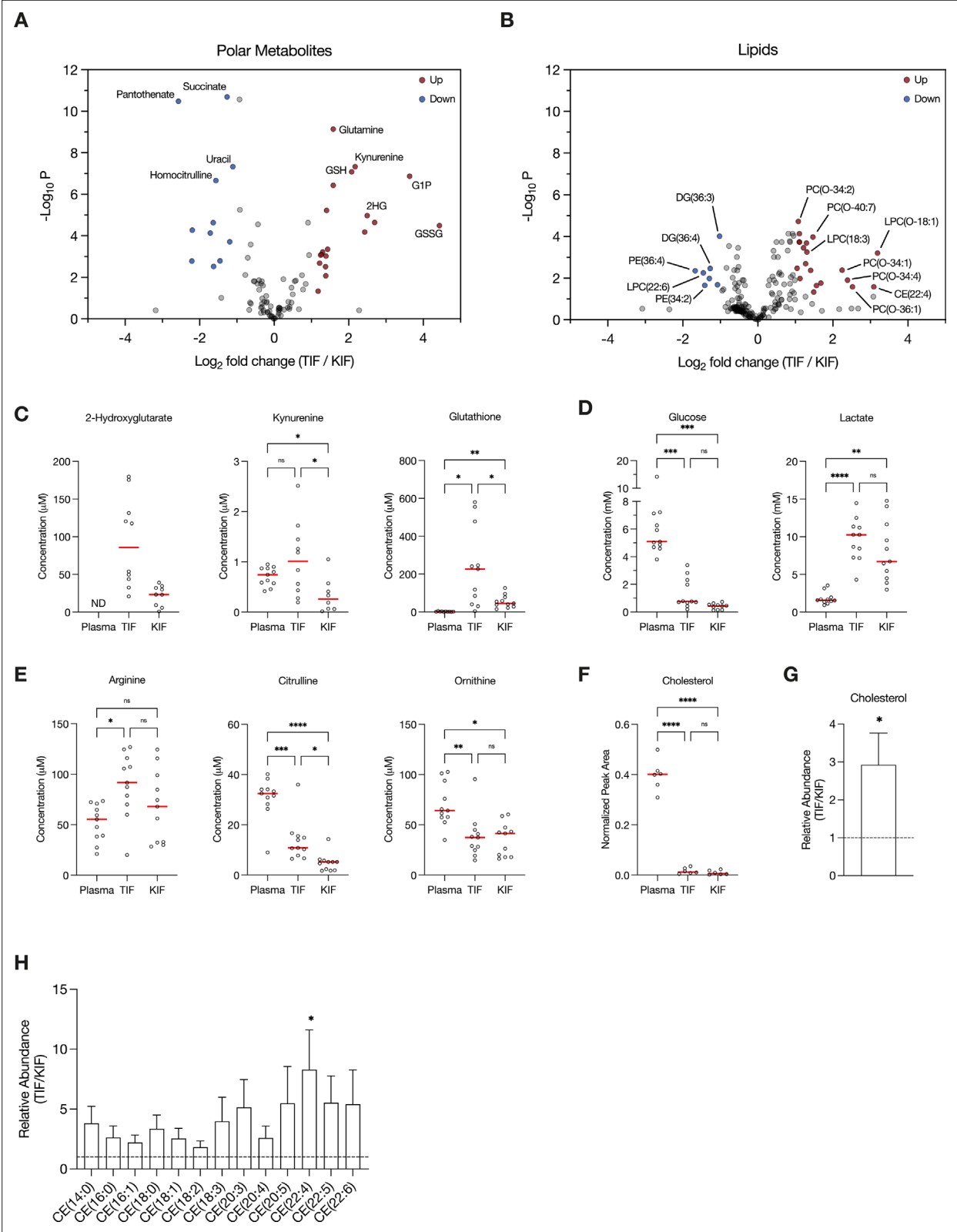

**Figure 2.** Assessment of metabolites that differ between renal cell carcinoma (RCC) interstitial fluid and normal kidney interstitial fluid (KIF). Volcano plots depicting the $\log_2$ fold change in polar metabolite concentration (**A**) or relative lipid levels (**B**) between tumor interstitial fluid (TIF) and KIF from RCC patients ($n$ = 55 patients in [A], $n$ = 38 patients in [B]). Cutoffs of |$\log_2$ fold change| >1 and adjusted p-value (false discovery rate-corrected) <0.05 were used to select significantly altered metabolites. Metabolites or lipids highlighted in red and blue are significantly higher and lower in TIF compared

*Figure 2 continued on next page*

*Figure 2 continued*

to KIF, respectively. Full names of selected lipids: PC(O-34:2), PC(P-34:1)/PC(O-34:2); PC(O-40:7), PC(P-40:6)/PC(O-40:7); LPC(O-18:1), LPC(P-18:0)/LPC(O-18:1); PC(O-34:1), PC(P-34:0)/PC(O-34:1); PC(O-34:4), PC(P-33:3)/PC(O-34:4); PC(O-36:1), PC(P-36:0)/PC(O-36:1). (**C–E**) Levels of selected metabolites measured by liquid chromatography/mass spectrometry (LC/MS) in plasma, TIF, and KIF from matched RCC patients (*n* = 10 patients). Each point represents a value measured from one patient, and the red line represents the mean across all patients considered. p-values were derived from either mixed-effects analysis (kynurenine, glutathione, glucose) or repeated measures one-way analysis of variance (ANOVA) (2-hydroxyglutarate, lactate, arginine, citrulline, ornithine), depending on whether missing values were present, and were Tukey multiple comparisons corrected (ns, not significant; *p < 0.05; **p < 0.01; ***p < 0.001; ****p < 0.0001). (**F**) Normalized peak area values of cholesterol measured by LC/MS in plasma, TIF, and KIF from matched RCC patients (*n* = 6 patients). Each point represents a sample, and the red line represents the mean across all patients considered. p-values were derived from repeated measures one-way ANOVA with Tukey multiple comparisons correction (ns, not significant; ****p < 0.0001). Relative abundance of cholesterol (**G**) or cholesteryl esters (**H**) in TIF compared to KIF from matched RCC patients (*n* = 20 patients). The mean is presented ± standard error of the mean (SEM), and the black dotted line indicates a ratio of 1, representing no difference in lipid levels between TIF and KIF. p-values were derived from a one sample *t*-test compared to 1 (*p < 0.05).

The online version of this article includes the following figure supplement(s) for figure 2:

**Figure supplement 1.** Heatmaps of metabolites that differ between renal cell carcinoma (RCC) interstitial fluid and normal kidney interstitial fluid (KIF).

**Figure supplement 2.** Assessment of metabolites that differ between plasma in patients with renal cell carcinoma (RCC) with plasma from normal individuals and from patients with non-small cell lung cancer (NSCLC).

**Figure supplement 3.** Plasma cystine concentration is affected by fasting.

compared to plasma, we did detect a roughly threefold elevation in cholesterol and cholesteryl esters in TIF compared to KIF (*Figure 2G,H*), which is intriguing given work showing ccRCC suppresses de novo cholesterol biosynthesis leading to a dependence on exogenous cholesterol import to meet metabolic demands (*Riscal et al., 2021*). These findings may argue that alterations in local lipid metabolism may be a feature of RCC to supply cancer cells with elevated cholesterol in the TME and raises questions about the dynamic regulation of cholesterol metabolism involving non-cancer cells in these tumors.

Lastly, we examined the composition of plasma from the RCC patients undergoing nephrectomy, and compared the polar metabolite levels to polar metabolite levels in plasma collected from healthy adult donors, and to polar metabolites levels in plasma from non-small cell lung cancer (NSCLC) patients (*Figure 2—figure supplement 2A–D*). Among the metabolites analyzed, intermediates in nucleotide metabolism stood out as being different between samples. Cystine concentration was also noted to be approximately twofold higher in plasma from patients with RCC compared to levels in plasma from healthy individuals based on the measurements here and available data from the Human Metabolome Database (HMDB; *Wishart et al., 2022*; *Figure 2—figure supplement 3A*). This finding demonstrates that there are influences from the tumor on the composition of nutrients in the circulating plasma. Although the RCC patients were thought to have localized disease, whether this influence is due to metabolic demands or contributions within the TME from circulating tumor cells, or as a consequence of undiagnosed metastasis, cannot be determined. The cystine finding was specifically notable because cystine levels have been demonstrated to influence sensitivity to glutaminase inhibitors (*Muir et al., 2017*), and RCC was a tumor type that showed early responses in glutaminase inhibitor clinical trials (*Lee et al., 2022*). The concentration of cystine in RCC plasma was comparable to that of NSCLC plasma, which suggests this may be a more common feature of tumor physiology than previously appreciated.

Since the plasma samples collected from both the RCC and NSCLC patients were collected following an overnight fast prior to surgery, whereas the healthy donor samples were collected from non-fasted donors, we considered the possibility that the fasting status of patients during the collection process may impact the concentration of circulating cystine. To explore this further, we collected fed and fasted plasma from a healthy donor and found a 1.6-fold increase in circulating cystine in the fasted state compared to the fed state (*Figure 2—figure supplement 3B*). This observation offers a plausible explanation for at least a component of the higher cystine concentration in RCC or NSCLC plasma, but a larger trial, including plasma from non-fasted cancer patients, would be needed to verify whether cystine levels are affected by fasting status. Moreover, the observed rise in cystine concentration due to fasting raises an intriguing possibility that therapies influenced by nutrient levels, such as glutaminase inhibitors (*Muir et al., 2017*), may also be influenced in part by a patient's diet or fed/fasted state (*Wilinski et al., 2019*; *LaBarre et al., 2021*), which could have effects on therapeutic

efficacy. Nevertheless, while considering the influence of fasting status on metabolite availability is important, other differences in plasma metabolite levels were unique to either RCC or NSCLC that cannot be explained by blood being collected in a fed or fasted state, such as the buildup of glutathione or uric acid in RCC compared to NSCLC plasma. Whether these other metabolite changes in blood have any biological or clinical significance will require further study.

Assessment of nutrient availability in other tumor types and tissues is needed to determine whether the similarity between nutrient availability in RCC and normal kidney tissue is a general feature of cancers, and examination of metastatic tumors compared to local tissue is also needed. Nevertheless, it is provocative to consider that general tissue homeostatic nutrient maintenance mechanisms are preserved in tumors, and that cancer cells may retain some dependence on nutrients defined by the adjacent origin tissue. To this end, it is notable that the metabolites found in RCC TIF differ from those reported in mouse pancreatic ductal adenocarcinoma (PDAC) TIF (*Sullivan et al., 2019*; *Apiz Saab et al., 2023*), and from those found in interstitial fluid derived from mouse brain and mammary fat pad tissue (*Ferraro et al., 2021*). Findings from analysis of PDAC TIF from tumors implanted in different tissue sites also noted differences in metabolite levels (*Sullivan et al., 2019*), supporting the notion that cancer must adapt to a specific tissue nutrient environment rather than determine the majority of nutrients present. These findings may suggest that each tissue represents a unique nutrient environment driven my interactions between resident non-cancer cell types, with potential implications for cancer metastasis. That is, this model raises the possibility that accessing a tissue nutrient environment that shares features with the primary cancer site is needed to support proliferation of metastatic cancer cells, and may contribute to the stereotyped pattern of metastases associated with cancers arising in specific tissues (*Riihimäki et al., 2018*). These findings may also present a challenge for therapies targeting nutrient dependencies in patients with metastatic tumors given that available metabolites could differ in each organ of metastasis.

We acknowledge some caveats with how to interpret these findings. Quantification of steady-state nutrient levels does not directly reflect nutrient use, as nutrient consumption and release rates cannot be derived from these steady-state measurements. An increase in levels of a metabolite in tissue relative to blood must represent some local production in the tissue, while a decrease in the levels of a metabolite relative to blood must represent some local consumption, however a lack of change does not mean a metabolite is not involved in metabolism in that tissue. Exchange of metabolites between blood and tissues may be faster than the rate of consumption or production, and we also did not consider differences between arterial and venous blood metabolites. Moreover, that KIF was collected from non-tumor renal tissue in specimens that were resected from patients with RCC raises the possibility that the levels of at least some of the metabolites measured in normal kidney were affected by the nearby tumor, or the fact that the patient had a cancer. Ideally, metabolites measured in KIF from healthy individuals would be compared with those measured in interstitial fluid from kidney tumors and adjacent normal tissues, although obtaining KIF from healthy patients was not possible for this study. Additionally, we also note that analysis of interstitial fluid will not take into account local variation in nutrient levels across different tissue regions (*Pan et al., 2016*; *Okegawa et al., 2017*; *Wang et al., 2022a*; *Miller et al., 2023*; *Wang et al., 2022b*). Microenvironmental differences will be averaged in the pooled collection method employed in this study. Known tumor heterogeneity, including the presence of both cancer and non-cancer cell types, are likely to drive gradients in nutrient availability within tumors (*Morandi et al., 2016*; *Lyssiotis and Kimmelman, 2017*). Lastly, the processing methodology employed to collect interstitial fluid may itself lead to some cell lysis and release of intracellular metabolites, which would affect the measured metabolite concentrations. Despite the limitations, this dataset provides a resource for future studies of RCC and normal kidney metabolism. The findings also emphasize that metabolites levels measured in one cancer type cannot necessarily be extrapolated to reflect which nutrients are found in the TME of other tumors. Ongoing efforts to understand how nutrient availability varies across different cancers may inform therapeutic strategies that take into account both tumor genetics and unique nutrient environments in tissues (*Abbott et al., 2023*).

# Methods

## Resource availability

### Lead contact

Further information and requests for resources and reagents should be directed to and will be fulfilled by the Lead Contacts, Jeffrey C. Rathmell (jeff.rathmell@vumc.org) and Matthew G. Vander Heiden (mvh@mit.edu).

## Materials availability

This study did not generate new unique reagents.

## Data and code availability

- Datasets can be found in *Supplementary files 1–3*.
- This paper does not report any original code.
- Any additional information required to reanalyze the data reported in this paper is available from the lead contacts upon request.

## Experimental model and study participant details

### Patient samples

Fresh RCC tumors and matched healthy tissue were surgically removed from 55 patients. *Supplementary file 1* contains relevant patient and tumor information. Samples were grossed by a trained pathologist in the Department of Pathology at Vanderbilt University Medical Center (VUMC). All pathological diagnoses were made by confirming gross specimen identity with histology. All metabolite samples were collected in accordance with the Declaration of Helsinki principles under a protocol approved by the VUMC Institutional Review Board (protocol no. 151549). Informed consent was received from all patients before inclusion in the study.

## Collection of TIF, KIF, and plasma from RCC patients

A 0.5–5 mg portion of histology-confirmed tumor and matched normal kidney tissue were placed in PBS on ice, then cut into smaller chunks to fit into the 0.22 µm nylon filter-containing Corning centrifuge tubes (Corning CLS8169). Tumor fragments were not minced, smashed, or dissociated in order to minimize sample manipulation and possible cell lysis. Both the normal kidney tissue and tumor were processed identically and side by side. Specimens were centrifuged at 4°C for 5 min at 300 × $g$ and the interstitial fluid-containing filtrate was collected. The tumor samples were not further processed. Plasma samples were collected as part of perioperative care into ethylenediaminetetraacetic acid (EDTA)-coated tubes, then centrifuged at 800 × $g$ for 5 min at 4°C; supernatants were further centrifuged at 3000 × $g$ for 20 min at 4°C to remove platelets. All samples were snap-frozen and stored at −80°C prior to analysis by liquid chromatography/mass spectrometry (LC/MS).

Tissue interstitial fluid was collected from freshly resected RCC tumors and matched healthy kidney tissue. Specimens were centrifuged against a 0.22-µm nylon filter (Corning CLS8169) at 4°C for 5 min at 300 × $g$. Plasma samples were collected as part of perioperative care into EDTA-coated tubes, then centrifuged at 800 × $g$ for 5 min at 4°C; supernatants were further centrifuged at 3000 × $g$ for 20 min at 4°C to remove platelets. All samples were snap-frozen and stored at −80°C prior to analysis by LC/MS.

## Collection of plasma from healthy adults

Non-fasting blood samples from ten healthy adult volunteers (four female and six male, ages ranging from 22 to 40 years) were collected and processed as described previously (*Abbott et al., 2023*). Briefly, plasma was collected using 21 G needles into 4-ml EDTA Vacutainer tubes (BD, 367839), then centrifuged at 800 × $g$ for 5 min at 4°C to remove cells. Supernatants were then further centrifuged at 3000 × $g$ for 20 min at 4°C to remove platelets. Samples were snap-frozen and stored at −80°C prior to analysis by LC/MS. The time between collection and processing of each sample was <10 min. Ethical approval for the collection of plasma was granted by the University of Cambridge Human Biology Research Ethics Committee (ref. HBREC.17.20).

## Collection of plasma from NSCLC patients

At time of oncologic resection, blood samples from 20 NSCLC patients were partitioned into 1 ml aliquots in EDTA-coated cryovials before being centrifuged at 800 × $g$ for 10 min at 4°C. Supernatant was transferred into Eppendorf tubes, snap-frozen, and stored at −80°C prior to analysis by LC/MS. Ethical approval for the collection and analysis of human fluids was granted by the University of Chicago Medical Center (IRB: UCMC 20-1696).

## Collection of fed and fasted plasma

Following an overnight fast of >9.5 hr by a volunteer (male, 50 years old), at 9:00 am 5 ml of fasted blood from the antecubital vein was collected into an EDTA Vacutainer tube (BD, 366643), then centrifuged at 800 × $g$ for 10 min at 4°C. Supernatant was transferred into Eppendorf tubes, snap-frozen, and stored at −80°C prior to analysis by LC/MS. Fed blood was collected on the same day at 5 pm, following a meal at 12:30 pm. Fed blood was processed in the same manner as the fasted blood.

## Method details

### Metabolite analyses

#### Quantification of metabolite levels in biological fluids

Metabolite quantification in human fluid samples was performed as described previously (*Sullivan et al., 2019*). 5 µl of sample or external chemical standard pool (ranging from ~5 mM to ~1 µM) was mixed with 45 µl of acetonitrile:methanol:formic acid (75:25:0.1) extraction mix including isotopically labeled internal standards (see materials section). All solvents used in the extraction mix were high-performance liquid chromatography (HPLC) grade. Samples were vortexed for 15 min at 4°C and insoluble material was sedimented by centrifugation at 16,000 × $g$ for 10 min at 4°C. 20 µl of the soluble polar metabolite extract was taken for LC/MS analysis. After LC/MS analysis, metabolite identification was performed with XCalibur 2.2 software (Thermo Fisher Scientific, Waltham, MA) using a 5-ppm mass accuracy and a 0.5-min retention time window. For metabolite identification, external standard pools were used for assignment of metabolites to peaks at given *m/z* and retention time, and to determine the limit of detection for each metabolite, which ranged from 100 nM to 3 µM (see *Supplementary file 2* for the *m/z* and retention time for each metabolite analyzed). After metabolite identification, quantification was performed by two separate methods for either quantification by stable isotope dilution or external standard. For quantification by stable isotope dilution, where internal standards were available, we first compared the peak areas of the stable isotope-labeled internal standards with the external standard pools diluted at known concentrations. This allowed for quantification of the concentration of labeled internal standards in the extraction mix. Subsequently, we compared the peak area of a given unlabeled metabolite in each sample with the peak area of the now quantified internal standard to determine the concentration of that metabolite in the sample. 51 metabolites were quantitated using this internal standard method (see *Supplementary file 2* for the metabolites quantitated with internal standards). For metabolites without internal standards, quantification by external calibration was performed as described below. First, the peak area of each externally calibrated analyte was normalized to the peak area of a labeled amino acid internal standard that eluted at roughly the same retention time to account for differences in recovery between samples (see *Supplementary file 2* for the labeled amino acid paired to each metabolite analyzed without an internal standard). This normalization was performed in both biological samples and external standard pool dilutions. From the normalized peak areas of metabolites in the external standard pool dilutions, we generated a standard curve describing the relationship between metabolite concentration and normalized peak area. The standard curves were linear with fits typically at or above $r^2 = 0.95$. Metabolites which did not meet these criteria were excluded from further analysis. These equations were then used to convert normalized peak areas of analytes in each sample into analyte concentration in the samples. Fifty-three metabolites were quantitated using this method.

#### LC/MS analysis

Metabolite profiling was conducted on a QExactive bench top orbitrap mass spectrometer equipped with an Ion Max source and a HESI II probe, which was coupled to a Dionex UltiMate 3000 HPLC system (Thermo Fisher Scientific, San Jose, CA). External mass calibration was performed using the standard calibration mixture every 7 days. An additional custom mass calibration was performed

weekly alongside standard mass calibrations to calibrate the lower end of the spectrum ($m/z$ 70–1050 positive mode and $m/z$ 60–900 negative mode) using the standard calibration mixtures spiked with glycine (positive mode) and aspartate (negative mode). 2 µl of each sample was injected onto a SeQuant ZIC-pHILIC 150 × 2.1 mm analytical column equipped with a 2.1 × 20 mm guard column (both 5 mm particle size; EMD Millipore). Buffer A was 20 mM ammonium carbonate, 0.1% ammonium hydroxide; Buffer B was acetonitrile. The column oven and autosampler tray were held at 25 and 4°C, respectively. The chromatographic gradient was run at a flow rate of 0.150 ml min$^{-1}$ as follows: 0–20 min: linear gradient from 80% to 20% B; 20–20.5 min: linear gradient form 20% to 80% B; 20.5–28 min: hold at 80% B. The mass spectrometer was operated in full-scan, polarity-switching mode, with the spray voltage set to 3.0 kV, the heated capillary held at 275°C, and the HESI probe held at 350°C. The sheath gas flow was set to 40 units, the auxiliary gas flow was set to 15 units, and the sweep gas flow was set to 1 unit. MS data acquisition was performed in a range of $m/z$ = 70–1000, with the resolution set at 70,000, the AGC target at $1 \times 10^6$, and the maximum injection time at 20 ms.

## LC–MS lipidomics

Positive ion mode analyses of polar and non-polar lipids were conducted using an LC–MS system composed of a Shimadzu Nexera X2 U-HPLC (Shimadzu) coupled to an Exactive Plus orbitrap mass spectrometer (ThermoFisher Scientific). 10 µl of human fluid sample was precipitated with 190 µl of isopropanol containing 1,2-didodecanoyl-sn-glycero-3-phosphocholine (Avanti Polar Lipids) as an internal standard. After centrifugation, 2 µl of supernatant was injected directly onto a 100 × 2.1 mm, 1.7 µm ACQUITY BEH C8 column (Waters). The column was eluted isocratically with 80% mobile phase A (95:5:0.1 vol/vol/vol 10 mM ammonium acetate/methanol/formic acid) for 1 min followed by a linear gradient to 80% mobile phase B (99.9:0.1 vol/vol methanol/ formic acid) over 2 min, a linear gradient to 100% mobile phase B over 7 min, then 3 min at 100% mobile phase B. Mass spectrometry analyses were performed using electrospray ionization in the positive ion mode using full scan analysis over 220–1100 $m/z$ at 70,000 resolution and 3 Hz data acquisition rate. Other mass spectrometry settings were as follows: sheath gas 50, in source collision-induced dissociation 5 eV, sweep gas 5, spray voltage 3 kV, capillary temperature 300°C, S-lens RF 60, heater temperature 300°C, microscans 1, automatic gain control target $1 \times 10^6$, and maximum ion time 100 ms. Lipid identities were determined on the basis of comparison to reference standards and reference plasma extracts and were denoted by the total number of carbons in the lipid acyl chain(s) and total number of double bonds in the lipid acyl chain(s). Ion counts normalized to an internal standard were reported, and are presented in *Supplementary file 3*.

## Quantification and statistical analysis

After determining the concentration of each metabolite in each plasma or interstitial fluid sample, all multivariate statistical analysis on the data was performed using R version 4.1.2 (2021-11-01). (*R Development Core Team, 2021*). For clustering, we used *factoextra* package for the analysis (*Kassambara and Mundt, 2020*). Metabolite concentrations were scaled by sum and metabolites that contained greater than 50% missing values were removed prior to analysis. For the remainder of the metabolites, missing values were replaced using 1/5th of the lowest positive value. Univariate analysis was performed comparing metabolite levels between groups (*t*-test) where significantly altered metabolites were defined by |log$_2$ fold change| >1 and adjusted p-value (Benjamini–Hochberg false discovery rate-corrected) <0.05 and assuming unequal group variance. Using the log$_2$ fold change and adjusted p-value cutoffs, the number of differentially expressed metabolites was determined. To determine the variability between the comparisons, chi-square test was performed using *chisq.test* function from *stats* package in R. All further statistical information is described in the figure legends. Owing to challenges in the sampling process, not all patients provided matched plasma, TIF, and KIF samples. Consequently, *Figure 2C–F* exclusively features data from patients where all three sites were sampled. For *Figure 2G,H*, matched TIF and KIF values were utilized. In all other plots throughout the study, patient samples were unpaired.

# Acknowledgements

We thank members of the Vander Heiden laboratory for helpful discussions. KLA was supported by the National Science Foundation (DGE-1122374) and National Institutes of Health (NIH) (F31CA271787,

T32GM007287). AA received support as a Howard Hughes Medical Institute (HHMI) Medical Research Fellow. BIR was supported by the NIH (F30CA247202, T32GM007347) and the American Association for Cancer Research (AACR). RKJ is supported by grants from the NIH (U01-CA224348; R01-CA259253; R01-CA208205; R01-NS118929; U01CA261842), the Ludwig Cancer Center at Harvard, Nile Albright Research Foundation, National Foundation for Cancer Research, and Jane's Trust Foundation. WKR acknowledges support from AACR and WKR and JCR acknowledge the Department of Defense (W81XWH-22-1-0418) and NIH (R01CA217987). MGVH acknowledges support from the MIT Center for Precision Cancer Medicine, the Ludwig Center at MIT, and the NIH (R35CA242379, P30CA1405141). The metabolomics work performed in NSCLC patients was supported by the Lung Cancer Research Foundation.

## Additional information

### Competing interests

Christopher S Nabel: Royalty and income from Cambridge Epigenetix and ThermoFisher and stock in Opko Health. Rakesh K Jain: Consultant/SAB fees from Cur, DynamiCure, Elpis, SPARC, SynDevRx; owns equity in Accurius, Enlight, SynDevRx; served on the Board of Trustees of Tekla Healthcare Investors, Tekla Life Sciences Investors, Tekla Healthcare Opportunities Fund, Tekla World Healthcare Fund, and received Research Grants from Boehringer Ingelheim and Sanofi; no funding or reagents from these organizations were used in the study. Matthew G Vander Heiden: Scientific advisor for Agios Pharmaceuticals, iTeos Therapeutics, Sage Therapeutics, Pretzel Therapeutics, Lime Therapeutics, Faeth Therapeutics, Droia Ventures, and Auron Therapeutics. The other authors declare that no competing interests exist.

### Funding

| Funder | Grant reference number | Author |
|---|---|---|
| National Science Foundation | DGE-1122374 | Keene L Abbott |
| National Cancer Institute | F31CA271787 | Keene L Abbott |
| National Institutes of Health | T32GM007287 | Keene L Abbott |
| Howard Hughes Medical Institute | Medical Research Fellow | Ahmed Ali |
| National Cancer Institute | F30CA247202 | Bradley I Reinfeld |
| National Institutes of Health | T32GM007347 | Bradley I Reinfeld |
| American Association for Cancer Research | | Bradley I Reinfeld |
| National Institutes of Health | U01-CA224348 | Rakesh K Jain |
| National Institutes of Health | R01-CA259253 | Rakesh K Jain |
| National Institutes of Health | R01-CA208205 | Rakesh K Jain |
| National Institutes of Health | R01-NS118929 | Rakesh K Jain |
| National Institutes of Health | U01CA261842 | Rakesh K Jain |
| Ludwig Cancer Center at Harvard | | Rakesh K Jain |

| Funder | Grant reference number | Author |
|---|---|---|
| Nile Albright Research Foundation | | Rakesh K Jain |
| National Foundation for Cancer Research | | Rakesh K Jain |
| Jane's Trust Foundation | | Rakesh K Jain |
| American Association for Cancer Research | | W Kimryn Rathmell |
| Department of Defense | Education Activity W81XWH-22-1-0418 | W Kimryn Rathmell Jeffrey Rathmell |
| National Cancer Institute | R01CA217987 | W Kimryn Rathmell Jeffrey Rathmell |
| MIT Center for Precision Cancer Medicine | | Matthew G Vander Heiden |
| Ludwig Center at MIT | | Matthew G Vander Heiden |
| National Cancer Institute | R35CA242379 | Matthew G Vander Heiden |
| National Cancer Institute | P30CA1405141 | Matthew G Vander Heiden |

The funders had no role in study design, data collection, and interpretation, or the decision to submit the work for publication.

## Author contributions
Keene L Abbott, Ahmed Ali, Bradley I Reinfeld, Conceptualization, Investigation, Methodology, Writing – original draft, Writing – review and editing; Amy Deik, Sonu Subudhi, Madelyn D Landis, Rachel A Hongo, Kirsten L Young, Tenzin Kunchok, Christopher S Nabel, Kayla D Crowder, Johnathan R Kent, Investigation, Writing – review and editing; Maria Lucia L Madariaga, Rakesh K Jain, Kathryn E Beckermann, Caroline A Lewis, Clary B Clish, Resources, Writing – review and editing; Alexander Muir, Resources, Methodology, Writing – review and editing; W Kimryn Rathmell, Jeffrey Rathmell, Matthew G Vander Heiden, Conceptualization, Resources, Supervision, Funding acquisition, Methodology, Writing – original draft, Writing – review and editing

## Author ORCIDs
Keene L Abbott ⓘ http://orcid.org/0000-0002-6166-704X
Amy Deik ⓘ http://orcid.org/0000-0002-9687-0953
Sonu Subudhi ⓘ http://orcid.org/0000-0002-5937-1880
Kayla D Crowder ⓘ http://orcid.org/0000-0002-0861-6489
Alexander Muir ⓘ https://orcid.org/0000-0003-3811-3054
Jeffrey Rathmell ⓘ http://orcid.org/0000-0002-4106-3396
Matthew G Vander Heiden ⓘ https://orcid.org/0000-0002-6702-4192

## Ethics
All metabolite samples were collected in accordance with the Declaration of Helsinki principles under a protocol approved by the VUMC Institutional Review Board (protocol no. 151549). Informed consent was received from all patients before inclusion in the study. Ethical approval for the collection and analysis of human fluids was granted by the University of Chicago Medical Center (IRB: UCMC 20-1696). Ethical approval for the collection of plasma was granted by the University of Cambridge Human Biology Research Ethics Committee (ref. HBREC.17.20).

Reviewer #1 (Public review): https://doi.org/10.7554/eLife.95652.3.sa1
Reviewer #2 (Public review): https://doi.org/10.7554/eLife.95652.3.sa2
Reviewer #3 (Public review): https://doi.org/10.7554/eLife.95652.3.sa3
Author response https://doi.org/10.7554/eLife.95652.3.sa4

## Additional files

### Supplementary files
• Supplementary file 1. Patient sample information.
• Supplementary file 2. Polar metabolite concentrations and related information.
• Supplementary file 3. Lipid concentrations and related information.
• MDAR checklist

### Data availability
All data generated or analyzed during the student are included with the manuscript and supporting files.

The following previously published dataset was used:

| Author(s) | Year | Dataset title | Dataset URL | Database and Identifier |
|---|---|---|---|---|
| Wishart DS, Guo A, Oler E, Wang F, Anjum A, Peters H, Dizon R, Sayeeda Z, Tian S, Lee BL, Berjanskii M, Mah R, Yamamoto M, Jovel J, Torres-Calzada C, Hiebert-Giesbrecht M, Lui VW, Varshavi D, Varshavi D, Allen D, Arndt D, Khetarpal N, Sivakumaran A, Harford K, Sanford S, Yee K, Cao X, Budinski Z, Liigand J, Zhang L, Zheng J, Mandal R, Karu N, Dambrova M, Schiöth HB, Greiner R, Gautam V | 2022 | HMDB 5.0: the Human Metabolome Database for 2022 | http://www.hmdb.ca/ | HMDB5.0, HMDB 5.0 |

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
