## [Editor Report · eLife assessment]

This study provides an **important** finding that the local abundance of metabolites impacts the biology of the tumor microenvironment by utilizing kidney tumors from patients and adjacent normal tissues. The evidence supporting the claims of the authors is **convincing**. The work will be of interest to the research community working on metabolism and kidney cancer especially.

---

## [Referee Report · Reviewer #1 (Public review)]

(a) Summary: The present study addresses how the local abundance of metabolites impacts the biology of the tumor microenvironment. The authors enroll patients harboring kidney tumors and use freshly resected tumor material for metabolic studies. Specifically, the authors separate the adjacent normal kidney tissue from the tumor material and then harvest the interstitial fluid from the normal kidney (KIF) or the tumor (TIF) for quantitative metabolomics. The plasma samples from the patient are used for comparison. Additionally, the authors also compare metabolite levels in the plasma of patients with kidney versus lung cancer (or healthy donors) to address how specific tumor types might contribute to circulating levels of metabolites. Altogether, the authors find that the metabolite levels in the KIF and TIF, although vastly different than plasma, are largely overlapping. These findings indicate that tissue of origin appears to have a stronger role in determining the local metabolic environment of tumors than the genetics or biochemistry of the tumor itself.

(b) Strengths: The biggest strength of the current study is the use of human patient-derived samples. The cohort size (~50 patients) is relatively large, which adds to the rigor of the work. The work also relies on a small pool of metabolites that can be quantitatively measured using methods developed by the authors. Focusing on a smaller metabolic pool also likely increases the signal-to-noise ratio and enables the more rigorous determination of any underlying differences. The manuscript is well-written and highlights both the significance of the findings and also acknowledges many of the caveats. The recognition of the metabolic contributions of surrounding normal tissue as the primary driver of local nutrient abundance is a novel finding in the work, which can be leveraged in future studies.

(c) Weaknesses: The work has certain caveats, some of which have been already recognized by the authors. These include the use of steady-state metabolites and the possibility of cross-contamination of some TIF into the adjacent KIF. This study is also unable to distinguish the mechanisms driving the metabolic changes in KIF/TIF relative to circulating levels in plasma.

The relative similarity of KIF and TIF is quite surprising. However, this interpretation is presently based on sampling of only ~100 polar metabolites and ~200 lipid molecules. It is, perhaps, possible that future technological developments that enable more comprehensive quantitative metabolic profiling might distinguish between KIF and TIF composition.

In vitro tissue culture is recognized to suffer from 'non-physiological' nutrient dependencies, which are impacted by the composition of culture media. Thus, in vivo studies remain our current gold-standard in mechanistic studies of tumor metabolism. It is presently unclear whether the findings of this work will be recapitulated in any of the kidney cancer in vivo models and thus be functionally testable.

The authors have acknowledged these caveats and where possible provided textual clarifications and updated figures in their revised manuscript. Future work will be required to model these changes in animal models.

---

## [Referee Report · Reviewer #2 (Public review)]

The study employs quantitative metabolomic and lipidomic analyses to scrutinize tumor interstitial fluid (TIF), adjacent normal kidney interstitial fluid (KIF), and plasma samples from renal cell carcinoma (RCC) patients. The authors delve into the intricate world of renal cell carcinoma and its tumor microenvironment, shedding light on the factors that shape nutrient availability in both cancerous and adjacent normal tissues. The authors prove that non-cancer-driven tissue factors play a dominant role in shaping nutrient availability in RCC. This finding opens up new avenues for research, suggesting that the tumor microenvironment is profoundly influenced by factors beyond the presence of cancer cells. This study not only contributes valuable insights into RCC metabolism but also prompts a reevaluation of the factors governing nutrient availability in tumor microenvironments more broadly. Overall, it represents a significant step forward in our understanding of the intricate interplay between cancer and its surrounding milieu.

The study is overall well-constructed, including appropriate analysis. Likewise, the manuscript is written clearly and supported by high-quality figures. Since the authors exclusively employed samples from RCC patients and did not include kidney interstitial fluid and plasma samples from healthy individuals, we cannot accurately assess the true significance and applicability of the results until the role of cancer cells in reshaping KIF is understood. In essence, some metabolite levels in the tumor interstitial fluid did not show an increase or decrease compared to the adjacent normal kidney interstitial fluid. However, the levels of these metabolites in both TIF and KIF might be higher or lower than those in kidney interstitial fluid from healthy individuals, and the roles of these metabolites should not be overlooked. Similar concerns extend to plasma levels, emphasizing the importance of metabolites that synchronously change in RCC TIF, KIF, and plasma-whether elevated or reduced.

---

## [Referee Report · Reviewer #3 (Public review)]

In this study, the authors utilized mass spectrometry-based quantification of polar metabolites and lipids in normal and cancerous tissue interstitial fluid and plasma. This showed that nutrient availability in tumor interstitial fluid was similar to that of interstitial fluid in adjacent normal kidney tissue, but that nutrients found in both interstitial fluid compartments were different from those found in plasma. This suggests that the nutrients in kidney tissue differ from those found in blood and that nutrients found in kidney tumors are largely dictated by factors shared with normal kidney tissue. Those data could be useful as a resource to support further study and modeling of the local environment of RCC and normal kidney physiology.

---

## [Author Response]

The following is the authors’ response to the original reviews.

**eLife assessment**
This study provides an important finding that the local abundance of metabolites impacts the biology of the tumor microenvironment by utilizing kidney tumors from patients and adjacent normal tissues. The evidence supporting the claims of the authors is convincing although certain caveats need to be taken into consideration as the authors acknowledged in the paper. The work will be of interest to the research community working on metabolism and on kidney cancer especially.
**Public Reviews:**

**Reviewer #1 (Public Review):**
Summary:The present study addresses how the local abundance of metabolites impacts the biology of the tumor microenvironment. The authors enroll patients harboring kidney tumors and use freshly resected tumor material for metabolic studies. Specifically, the authors separate the adjacent normal kidney tissue from the tumor material and then harvest the interstitial fluid from the normal kidney (KIF) or the tumor (TIF) for quantitative metabolomics. The plasma samples from the patient are used for comparison. Additionally, the authors also compare metabolite levels in the plasma of patients with kidney versus lung cancer (or healthy donors) to address how specific tumor types might contribute to circulating levels of metabolites. Altogether, the authors find that the metabolite levels in the KIF and TIF, although vastly different than plasma, are largely overlapping. These findings indicate that tissue of origin appears to have a stronger role in determining the local metabolic environment of tumors than the genetics or biochemistry of the tumor itself.Strengths:The biggest strength of the current study is the use of human patient-derived samples. The cohort size (~50 patients) is relatively large, which adds to the rigor of the work. The work also relies on a small pool of metabolites that can be quantitatively measured using methods developed by the authors. Focusing on a smaller metabolic pool also likely increases the signal-to-noise ratio and enables the more rigorous determination of any underlying differences. The manuscript is well-written and highlights both the significance of the findings and also acknowledges many of the caveats. The recognition of the metabolic contributions of surrounding normal tissue as the primary driver of local nutrient abundance is a novel finding in the work, which can be leveraged in future studies.

We thank the Reviewer for their careful evaluation of the study and for their supportive comments.

Weaknesses:The work has certain caveats, some of which have been already recognized by the authors. These include the use of steady-state metabolites and the possibility of cross-contamination of some TIF into the adjacent KIF. This study is also unable to distinguish the mechanisms driving the metabolic changes in KIF/TIF relative to circulating levels in plasma.

We agree with the Reviewer that these are important caveats to consider when interpreting the results of this study.

The relative similarity of KIF and TIF is quite surprising. However, this interpretation is presently based on a sampling of only ~100 polar metabolites and ~200 lipid molecules. It is, perhaps, possible that future technological developments that enable more comprehensive quantitative metabolic profiling might distinguish between KIF and TIF composition.

The Reviewer raises another important point that our interpretation of KIF vs TIF is limited to the ~300 metabolites we measured. We agree it would be worthwhile quantifying more metabolites where technically feasible to further characterize similarities and differences in nutrient availability between tumor and normal tissues.

In vitro, tissue culture is recognized to suffer from ‘non-physiological’ nutrient dependencies, which are impacted by the composition of culture media. Thus, in vivo studies remain our current gold-standard in mechanistic studies of tumor metabolism. It is presently unclear whether the findings of this work will be recapitulated in any of the kidney cancer in vivo models and thus be functionally testable.

We thank the Reviewer for calling attention to the limitations of cell culture media in studying tumor metabolism. While both in vitro and in vivo approaches have inherent limitations, formulating culture media based on metabolite concentrations measured here and in other studies provides a tool to study the influence of nutrient availability on kidney cell or kidney cancer cell phenotypes in vitro. We also agree with the Reviewer that determining whether the findings in our study are recapitulated in mouse models of kidney cancer, as this might enable investigation into the factors that modulate nutrient availability in this tissue context.

**Reviewer #2 (Public Review):**
The study employs quantitative metabolomic and lipidomic analyses to scrutinize tumor interstitial fluid (TIF), adjacent normal kidney interstitial fluid (KIF), and plasma samples from renal cell carcinoma (RCC) patients. The authors delve into the intricate world of renal cell carcinoma and its tumor microenvironment, shedding light on the factors that shape nutrient availability in both cancerous and adjacent normal tissues. The authors prove that non-cancer-driven tissue factors play a dominant role in shaping nutrient availability in RCC. This finding opens up new avenues for research, suggesting that the tumor microenvironment is profoundly influenced by factors beyond the presence of cancer cells. This study not only contributes valuable insights into RCC metabolism but also prompts a reevaluation of the factors governing nutrient availability in tumor microenvironments more broadly. Overall, it represents a significant step forward in our understanding of the intricate interplay between cancer and its surrounding milieu.

We thank the Reviewer for their evaluation of our work and for their supportive comments.

The study is overall well-constructed, including appropriate analysis. Likewise, the manuscript is written clearly and supported by high-quality figures. Since the authors exclusively employed samples from RCC patients and did not include kidney interstitial fluid and plasma samples from healthy individuals, we cannot accurately assess the true significance and applicability of the results until the role of cancer cells in reshaping KIF is understood. In essence, some metabolite levels in the tumor interstitial fluid did not show an increase or decrease compared to the adjacent normal kidney interstitial fluid. However, the levels of these metabolites in both TIF and KIF might be higher or lower than those in kidney interstitial fluid from healthy individuals, and the roles of these metabolites should not be overlooked. Similar concerns extend to plasma levels, emphasizing the importance of metabolites that synchronously change in RCC TIF, KIF, and plasma-whether elevated or reduced.

We agree with the Reviewer that an important caveat in considering the study findings is that we do not have KIF values from healthy individuals. Since resection of normal kidney is not a common procedure, obtaining KIF samples from healthy patients was not possible to complement our analysis. We further agree that the metabolite levels we measured in KIF or plasma are plausibly impacted by the presence of RCC. We did compare the composition of polar metabolites in the plasma from RCC, lung cancer, and healthy patients, highlighting how cystine is affected by tumor presence and/or sample collection methodology. We also point out that factors such as diet will impact metabolites in both blood and tissues.

**Reviewer #3 (Public Review):**
In this study, the authors utilized mass spectrometry-based quantification of polar metabolites and lipids in normal and cancerous tissue interstitial fluid and plasma. This showed that nutrient availability in tumor interstitial fluid was similar to that of interstitial fluid in adjacent normal kidney tissue, but that nutrients found in both interstitial fluid compartments were different from those found in plasma. This suggests that the nutrients in kidney tissue differ from those found in blood and that nutrients found in kidney tumors are largely dictated by factors shared with normal kidney tissue. Those data could be useful as a resource to support further study and modeling of the local environment of RCC and normal kidney physiology.

We thank the Reviewer for their time considering our paper and for their supportive comments.

In Figures 1D and 1E, there were about 30% of polar metabolites and 25% of lipids significantly different between TIF and KIF, which could be key factors for RCC tumors. This reviewer considers that the authors should make comments on this.

We agree with the Reviewer that the metabolites that significantly differ between TIF and KIF are of interest, particularly for those studying RCC tumor metabolism. We comment on some of the metabolites driving differences between TIF and KIF in our discussion of Figure 2, and in the revised manuscript we now include a new figure showing a heatmap that enables visualization of these metabolites (Figure 2-Supplement 1A-B).

**Recommendations for the authors:**

**From the Reviewing Editor:**
Figure 2 needs to plot heatmaps for both upregulated and downregulated metabolites in TIF.

We agree and now include heatmaps for significantly differing polar metabolites and lipids in TIF vs KIF as requested by Reviewer 3 (Figure 2-Supplement 1A-B). For completeness, we also include heatmaps for metabolites differing between healthy and RCC plasma (Figure 2-Supplement 2C) and for NSCLC and RCC plasma (Figure 2-Supplement 2D).

There is a need to show whether the differences in these metabolites between plasma and tissue interstitial fluid are specific to RCC patients or if they are also present in normal individuals.

Unfortunately, it has not been possible for us to collect KIF from healthy individuals. Since resection of normal kidney is not a common procedure, we have no way to obtain sufficient KIF samples from healthy patients for this measurement. We discuss this as a limitation of the study.

**Reviewer #1 (Recommendations For The Authors):**
a. The authors should provide additional details about the methodology to separate the KIF and TIF. Contaminating metabolites from surrounding tissue or the peritoneal fluids could impact interpretation and it would be helpful to understand how these challenges were addressed during tissue collection for this study. Additionally, was the collected tissue minced or otherwise dissociated? If so, could these procedures cause tissue lysis and contaminate the KIF/TIF with intracellular components?

We thank the Reviewer for the suggestions to include more information about the sampling methodology. Care was taken to minimize cell lysis incurred by the processing methodology as tissues were not minced, smashed, nor dissociated, however there is still a possibility of some level of tissue lysis that is pre-existing or occurs during the isolation procedure. We note this caveat in the text (lines 218-220) and have updated the Methods with more details of the sampling and processing of the samples.

b. Although the authors focus on metabolites that are elevated in TIF (relative to KIF and plasma), it would be equally relevant to consider the converse. Metabolites that are reduced in TIF, either due to underproduction or overconsumption, could render the tumors auxotrophic for some critical dependencies and identify some novel metabolic vulnerabilities. In this regard, Figure 2 could have a heatmap of the top metabolites that are elevated and depleted specifically in the TIF.

We agree with the Reviewer it is useful to include heatmaps to better display the metabolites that significantly differ between TIF and KIF and now include these in Figure 2-Supplement 1A-B.

c. The future utilization of this knowledge would depend on our ability to model these differences. Would interstitial tissue from a normal mouse kidney or tumor-bearing mouse kidney recapitulate the same differences relative to mouse plasma?

We agree with the Reviewer that it would be worth determining whether the findings in our study are recapitulated in mouse models of kidney cancer, which would support future investigation into the factors that modulate nutrient availability. This is an interesting question, but we did not have access to endogenously arising models of RCC, which have been a limitation for the field, and comparison of normal mouse kidney metabolite data to human metabolite data is problematic for obvious reasons. Thus, we had no choice but to discuss this as a limitation of the study.

**Reviewer #2 (Recommendations For The Authors):**
In this study, Abbott et al. investigated the metabolic profile of renal cell carcinoma (RCC) by analyzing the tumor interstitial fluid (TIF), adjacent normal kidney interstitial fluid (KIF), and plasma samples from patients. The results indicate that nutrient composition in TIF closely resembles that of KIF, suggesting that tissue-specific factors, rather than tumor-driven alterations, have a more significant impact on nutrient levels. These findings are interesting. The study is overall well-constructed, including appropriate analysis, and the manuscript is written clearly and supported by high-quality figures. However, some issues are raised which if addressed, would strengthen the paper.

We thank the Reviewer for their suggestions to improve the paper.

The authors found a difference in the number of metabolites when comparing TIF or KIF lipid composition with plasma. The discoveries are intriguing; however, I am keen to understand whether the differences in these metabolites between plasma and tissue interstitial fluid are specific to RCC patients or if they are also present in normal individuals. I am particularly interested in identifying which metabolites could serve as potential diagnostic markers, intervention targets, or potentially reshape the tumor microenvironment. Because, even though some metabolite levels show no difference between TIF and KIF in RCC patients, I wonder if these metabolite levels in KIF increase or decrease compared to the interstitial fluid in healthy individuals. I am intrigued by the metabolites that simultaneously increase or decrease in both TIF and KIF compared to the kidney interstitial fluid in healthy individuals.

We agree with the Reviewer that it would be interesting to measure kidney interstitial fluid from healthy patients to be able to compare metabolites changing due to the presence of RCC tumor. As we discuss in response to the public review, this was not possible as we could not obtain material from healthy individuals for analysis. Nevertheless we agree it warrants future study if material were available.

The analysis conducted using plasma from healthy donors, as applauded by the author, is noteworthy. The author seems to have found that cystine levels do not differ between RCC patient plasma and tissue interstitial fluid. However, considering that in patient plasma, the cystine concentration is approximately two-fold higher than in plasma from healthy individuals, likely, cystine levels in patient tissue fluid have also increased nearly two-fold compared to levels in the interstitial fluid of normal kidney tissues. This finding aligns with the discovery of elevated GSH levels in cancer cells.

We agree with the Reviewer that a higher cystine concentration in RCC patient plasma and interstitial fluid is interesting, and also considered this in relationship to past findings including reports of elevated GSH levels in RCC. However, we think this observation is driven at least in part by the fasting status of the patients pre-surgery. This does not rule out some part being related to the presence of the tumor, as this would be consistent with elevated GSH levels as noted by the Reviewer. Future studies will be needed to further delineate the factors that impact elevated cystine levels in both interstitial fluid and plasma.

Some minor typos, such as "HIF1-driven" should be corrected.

We thank the Reviewer for pointing out this typo and we have corrected it in the revised manuscript.